# Frontiers of Hydroxyapatite Composites in Bionic Bone Tissue Engineering

**DOI:** 10.3390/ma15238475

**Published:** 2022-11-28

**Authors:** Jingcun Shi, Wufei Dai, Anand Gupta, Bingqing Zhang, Ziqian Wu, Yuhan Zhang, Lisha Pan, Lei Wang

**Affiliations:** 1Department of Oral and Maxillofacial Surgery—Head & Neck Oncology, Shanghai Ninth People’s Hospital, Shanghai Jiao Tong University School of Medicine, Shanghai 200011, China; 2College of Stomatology, Shanghai Jiao Tong University, Shanghai 200011, China; 3Shanghai Key Laboratory of Stomatology, Shanghai Research Institute of Stomatology, National Clinical Research Center for Oral Diseases, National Center for Stomatology, Shanghai 200011, China; 4Department of Plastic and Reconstructive Surgery, Shanghai Ninth People’s Hospital, Shanghai Jiao Tong University School of Medicine, Shanghai 200011, China; 5Shanghai Tissue Engineering Key Laboratory, Shanghai Research Institute of Plastic and Reconstructive Surgey, Shanghai 200011, China; 6Department of Dentistry, Government Medical College & Hospital, Chandigarh 160017, India

**Keywords:** hydroxyapatite, composites, compound scaffolds, tissue-engineered bone, bone defect repair

## Abstract

Bone defects caused by various factors may cause morphological and functional disorders that can seriously affect patient’s quality of life. Autologous bone grafting is morbid, involves numerous complications, and provides limited volume at donor site. Hence, tissue-engineered bone is a better alternative for repair of bone defects and for promoting a patient’s functional recovery. Besides good biocompatibility, scaffolding materials represented by hydroxyapatite (HA) composites in tissue-engineered bone also have strong ability to guide bone regeneration. The development of manufacturing technology and advances in material science have made HA composite scaffolding more closely related to the composition and mechanical properties of natural bone. The surface morphology and pore diameter of the scaffold material are more important for cell proliferation, differentiation, and nutrient exchange. The degradation rate of the composite scaffold should match the rate of osteogenesis, and the loading of cells/cytokine is beneficial to promote the formation of new bone. In conclusion, there is no doubt that a breakthrough has been made in composition, mechanical properties, and degradation of HA composites. Biomimetic tissue-engineered bone based on vascularization and innervation show a promising future.

## 1. Introduction

In repairing bone defects, autologous bone grafting has always been considered the gold standard for bone reconstruction [1]. However, autologous bone grafting can be too morbid for the patient, and the donor site is prone to various complications. In long segments of bone defects in particular, this disadvantage is more obvious [2]. Allogenic transplantation for repairing bone defects presents several issues, such as immune rejection and pathogen transmission [3]. On the contrary, tissue-engineered bone offers numerous advantages, such as minimal damage, no need to open a second surgery zone, and satisfying the demands of mechanical resistance by the recipient bone [4]. Bone tissue engineering involves transplantation of seed cells onto a scaffold with excellent biocompatibility and proper mechanical strength. Then, the cell-and-scaffold complex is implanted into the human bone defect, and various bioactive factors are added to support cell proliferation and osteogenesis to repair bone defects [5] (Figure 1).

A thorough understanding of the process of bone repair is crucial to reconstruct the bone defects. The healing process of a fracture consists of three stages: inflammatory phase, proliferative phase, and remodeling phase. The first stage that occurs immediately after the fracture is the formation of a hematoma. Subsequently, endothelial cells, fibroblasts, and osteoblasts gather at the fracture gap and form granulation tissue. Then, a fibrous callus is created, which can limit the movement of the fracture. Typically, the inflammatory phase lasts up to sevem days. During the fiber growth phase, blood vessels continue to grow inward, and osteoid and collagen fibers appear to form the callus. Then, immature braided bone emerges and gradually forms a callus, further increasing the stability of the fracture site. The third stage involves callus formation and mineralization, replacing the callus with mineralized bone and restoring its original properties through bone remodeling [6]. The stress fracture model in mice suggests that the healing process of bone injury follows a certain spatio-temporal sequence, namely, nerve growth (1–3 d), vascularization (7 d), ossification (14 d), and mineralization (56 d) [7]. Different imaging techniques show different stages of bone healing. MicroCT observation of the healing process of stress fractures in mice showed that brachiated bone formation was observed on day 7 after injury, and callus size reached its maximum on day 14. Callus remodeling and cortical remodeling were the most significant on the 56th day [7]. In humans, 2–3 weeks after a fracture, X-ray examination can see the fracture line. This may be due to the soft callus formation [6]. It takes at least three months for the fracture line to disappear; that is, to achieve radiographically demonstrated bone union.

Bone tissue engineering emphasizes three main components: scaffolds, seed cells, and growth factors [8]. Among them, scaffold materials play a major role as they are carriers of seed cells and growth factors. Hence, the selection of a suitable scaffold is the most critical part of bone tissue engineering [9] (Figure 2). According to their properties, scaffolds can be divided into allogeneic demineralized bone, bioceramics, metals, polymers, and composites [10]. Allogeneic demineralized bone has the most similar properties and mechanical strength to natural bone, but attention should be paid to its immune reactivity and the risk of disease transmission. Tantalum and titanium as representative metal materials have excellent mechanical properties and biocompatibility, often used in the treatment of arthroplasty, joint replacement, spinal fusion, and femoral head necrosis [11,12]. Naturally derived biomaterials, such as fibrin and collagen, are produced by living organisms and eventually degrade into carbon dioxide and water. They possess good biocompatibility, come from a wide range of sources, and involve minimal adverse immune reactions [13,14,15,16]. Ceramic products, especially bioactive ceramics such as calcium phosphate ceramics (CPC), hydroxyapatite (HA), and bioglass, have a strong bone-inducing response in addition to providing physical support [17] and are more conducive to mutual transfer with the surrounding environment [18]. However, the degradation of ceramic materials in vivo usually takes decades, and their high brittleness limits their application. Therefore, natural materials and ceramic materials are usually used to make composites with excellent properties [19].

HA is a primary inorganic component of natural bone and used commonly as a scaffold in artificial bone because of its good biological activity and biocompatibility [20]. Some researchers have used the MC3T3-E1 cell line for a biological assessment of commercial nano-hydroxyapatite (nHA) and found that average length of nanoparticles is around 20–40 nm. According to ISO 10993-5, nanoparticles have cyto-compatibility. In vitro micronuclei experiments demonstrated that nHA exists only in cytoplasm and extracellular space and does not penetrate the nucleus, indicating that there is no genotoxicity to cells [20]. However, HA shows low toughness and a slow rate of degradation, which does not match with the rate of bone repair [21]. HA-polymer composites have been prepared by utilizing the excellent toughness and suitable degradation rate of polymer materials. They can meet the ideal requirements of a scaffold and become a common material for bone tissue engineering [22,23]. Furthermore, with the advent of nano-molecular technology, nHA has also become an important scaffold material in bone tissue engineering [24,25]. However, both HA and nHA have a high elastic modulus and higher fragility, since a material less elastically deformed than natural bone when subject to force results in resorption of native bone [26]. Therefore, how to improve the performance of HA or nHA composite scaffolds has become a hot topic in bone tissue engineering.

## 2. Optimization and Improvement of HA Composite Scaffolds to Improve Osteogenic Properties

HA is a scaffold material widely used in bone tissue engineering due to its high mechanical strength and good biocompatibility. The development of nanoscience has also introduced nanotechnology into scaffold materials, promoting research on nHA. Particle size, crystalline morphology, and combining nHA with other materials promote the growth of osteoblasts and development of various biomedical materials [27]. However, HA also presents certain deficiencies as a scaffold material, such as incompatibility between the degradation rate and new bone formation and low porosity and plasticity of the scaffold. The cells have a poor ability to adhere and proliferate on HA. This leads to challenges in the progress of bone tissue engineering. In addition to the basic support function, the scaffold can also be surface-modified to improve its affinity for cells and cytokines [28].

### 2.1. Mimic the Composition and Mechanical Properties of Natural Bone

Bone is a natural composite material composed of organic matter (collagen) and minerals (calcium phosphate, especially HA). HA is responsible for the mechanical strength of bones, whereas the toughness and elasticity are attributed to collagen. The challenge for bone tissue engineering is to develop biomimetic composite scaffolds that balance biological and biomechanical properties [29] (Figure 3). Based on the understanding of the composition of natural bone and in-depth research, attempts are being made to prepare biomimetic materials from organic content and natural mineralized collagen [30].

Olszta et al. aligned nHA crystals along the axis of collagen fibers to mimic native bone nanostructures [31]. Some scholars also used a co-titration method of phosphate collagen (COL) solution and calcium hydroxide solution under the assistance of microwave. During in situ precipitation, collagen fibers and HA may form simultaneously. The final COL/HA biomimetic scaffold contained intra- and inter-fibrous HA [32], similar to the microstructure of natural bone. Zhou et al. developed a calcium phosphate (CaP)/Col/HA scaffold [33]. Porous CaP ceramic material simulates a porous bone structure; the second level network structure was prepared by vacuum infusion; and the tertiary HA layer was fabricated by biomimetic mineralization. This scaffold is similar in structure and composition to natural bone. The bionic scaffold formed new bone faster than a normal CaP scaffold in the dorsal muscle of the rabbit, suggesting that bionic scaffolding has improved bone induction capacity. Similarly, Cheng et al. made injectable biomimetic hydrogels using silk nanofibers (SNF) and nHA, which are similar in composition and structure to natural bones [34]. These breakthroughs can make scaffold-based bone tissue engineering safer and convenient, thus better meeting the needs of clinical application.

When the elastic modulus of tissue-engineered bone is similar to that of natural bone, the damage to autologous bone is minimal and the mechanical compatibility is the best. The compressive strength of the natural bone is 2 to 230 MPa and the elastic modulus is 0.05 to 30 GPa [35,36]. HA has sufficient mechanical strength, but its texture is brittle and lacks toughness, which leads to autologous bone resorption and limits the application of HA. The mechanical properties of the HA composites are shown in Table 1.

Chitosan (CS) and sodium alginate (SAL) are elastic and easy to degrade. Liao et al. [65] prepared nHA/SAL/CS composites by in situ synthesis with a compressive strength of 34.3 MPa. Some scholars constructed nHA/poly(lactic-co-glycolic acid) (PLGA)/COL scaffolds through multistage polymerization and characterized their mechanical properties. The tensile strength of the nHA/PLGA/collagen membrane was observed to be similar to that of human woven bone [66]. Over time, PLGA components gradually hydrolyze. The secreted matrix and minerals can maintain the force of the scaffolds when cultured with differentiated human mesenchymal stem cells (hMSCs). Apart from being composed of different types of materials, the ratio between materials is also very important for controlling the mechanical properties. By comparing the polymers of lactic acid (PLA)/HA composite scaffolds with different mass ratios (5:5, 4:6, 3:7, 2:8, and 1:9), it was observed that when the ratio is 8:2, the overall performance is the best in terms of a compression force and an elastic modulus of 34.1 MPa and 2.63 GPa, respectively [40]. Researchers often enhance the mechanical properties of HA by adding various polymers or monitoring the ratio of different materials, and also characterize the mechanical properties of the scaffold. Kaufman et al. produced 30 to 300 indentations on 22 polymer–ceramic composites using a nano-indentation technique [67]. A power law was used to adapt the initial unloading curve to compare the reduced elastic modulus of various materials, providing a new reference method for determining the mechanical properties of the scaffolds. Unlike metal materials, when HA composites are implanted in vivo, the mechanical strength also changes with the formation of new bone and degradation of scaffolds. Therefore, it is necessary to pay attention to the mechanical strength of the scaffold before implantation and to dynamic changes after implantation. By incorporating nanomaterials with inorganic properties, such as carbon nanotubes (CNT) [68], and by preparing biphasic calcium phosphate in different proportions with TCP, the mechanical properties of HA composites are similar to those of human cancellous bone.

### 2.2. Modify the Surface Morphology and Pore Size of the Material

Recent studies have shown that conventional factors may affect cellular biological properties. The morphology of the culture vessel’s surface also influences cellular growth and differentiation. The safe and effective induction of directional differentiation of BMSCs is a major task. HA discs were ground with sandpaper of various sizes. Multiple sets of surface morphology were obtained with roughness ranging from 0.2 to 1.65 μm and peak distances ranging from 89.7 to 18.6 μm. It was found that hBMSCs had the best directional adhesion and osteogenic differentiation on surfaces with an average roughness of 0.77–1.09 μm and an average peak distance of 53.9–39.3 μm [69]. Tan et al. used photolithography combined with reactive ion engraving to control the scaffold’s surface at the micro and nano scales [70]. By depositing nHA on silicone surfaces at micro and nano scales, the cells were observed to most closely resemble the morphology observed in vivo, and the cell shape was closely associated with cell growth, differentiation, and phenotypic expression. Kim et al. found that scaffolds with staggered orthogonal double-layer and alternative double-layered structures promoted cell proliferation and osteogenic differentiation [71]. Additionally, micro–nano-hybrid (the hybrid of nano-rod and micro-rod) surface topographies had better performance on osteogenic differentiation of ADSCs (adipose derived stem cells) than nanosheets and nanorods [72]. Hence, modifying the surface morphology of the scaffolds, such as roughness, micron, nano scale, or arrangement structure, will affect the properties of the cells and determine osteogenesis.

The pore size of natural cancellous bone is 300–600 μm and porosity is 75–85% [37,38]. Large pore size and porosity are conducive to nutrient delivery and metabolite excretion. A scaffold’s porosity also plays a major role in cell and cytokine penetration and inducement of bone formation. Numerous studies have shown that larger pore sizes and porosity contribute to the formation of new bone and blood vessels. In general, pore size above 50 μm facilitates the entry of cells and angiogenesis. One of the latest research studies focused on the preparation of porous scaffolding. Zhou et al. soaked melamine foam in HA/CS composite suspension until liquid filled the foam [73]. Then, the composite material was dried by vacuum oven. As the moisture evaporated, vacuum suction guided uniform voids to form and connect with each other. The aperture range of 90–130 μm was observed. The CS and HA composites were injected into the composite braid, and porous scaffolds were prepared by freeze-drying and cold atmospheric plasma techniques. HA provides macroscopic pore size of the scaffold from 80 to 180 μm, while the cold atmospheric plasma treatment made the microscopic pore size of the scaffold ≤10 μm and the porosity of the scaffold ≥80% [74]. Dou et al. developed a PLGA/nHA/Gel scaffold. This composite scaffold featured a large front hole size of >1100 μm and a side hole of >500 μm, providing sufficient open space and reliable overall support for the growth of cells and tissues [75]. In addition to the characterization of the scaffold aperture, some in vivo/in vitro experiments were also carried out. Interconnected pores facilitated cell growth in the interior of the material. Scholars have prepared Poly (L-L-lactic acid) (PLLA) and HA scaffolds of different pore sizes. Studies have found that scaffolds with a 500–600 μm pore size promoted more osteoblast adhesion than scaffolds with a small pore size (150–315 μm and 315–400 μm) [76]. Lee et al. investigated the bone regeneration potential of nanocomposite polydopamine/HA/COL/calcium silicate scaffolds and implanted them into a rat bone defect model, finding higher new bone formation in the 500 μm pore size group than in the 250 μm group [77]. The porosity of the scaffold allows tissue growth and nutrition transfer, a finding that elicited widespread concern among researchers. Because porous structures are easily damaged at high temperatures, pore size, porosity, and mechanical strength of the scaffolds should be balanced to create load-bearing tissue-engineered bone. It is difficult to prepare a scaffold with both a porous structure and excellent mechanical properties. Therefore, it is necessary to optimize the creation process of the scaffold or to employ low temperature disinfection to tackle this problem.

### 2.3. Regulation of the Degradation Rate of Scaffolds

HA is a common scaffold used for bone tissue engineering and has good biocompatibility. However, the degradation rate of HA is slow, and it may last for several years. This mismatch between the degradation of the scaffold and the formation of bone seriously affects the quality of the tissue-engineered bone.

Bioactive glass is commonly used in bone tissue engineering and often used to enhance the mechanical properties. Zhang et al. prepared HA-CaO ceramics and found that the addition of CaO made the scaffold more easily degradable. They soaked the HA-CaO ceramic in Tris-buffered saline solution for 28 days. Due to the dissolution of CaO, it converted into CaCO_3_. The area fraction and the size and depth of pits for composite ceramics have increased compared to pure HA ceramics [78]. Shuai et al. [79] added poly (glycolic acid) (PGA) to HA/PLLA scaffolds to accelerate degradation. After a 28-day immersion, the weight loss rate increased from 3.3% to 25.0%. Since PGA is highly hydrophilic, it degrades first and accelerates PLPA hydrolysis. Scaffold degradation promotes both exposure and deposition of HA. The scaffold was implanted into the radius bone defect of a rabbit, and the defect area was repaired after eight weeks of implantation. In addition to studying the degradation of scaffolds in solution, there are scholars who studied biological degradation under the action of cells. Some scholars have prepared Li/HA scaffolds and studied their in vitro degradation by solution and osteoblast pathways. Studies have shown that the mechanism of simulated body fluid (SBF) degradation is significantly different from that of cells. In SBF, HA is deposited on the surface after hydrolysis of the Li/HA scaffold. The degradation of the scaffold by osteoblasts is a biological phenomenon [80]. Chang et al. reported HA/CS scaffolds where the degradation of compound materials was divided into a faster (about 10 wt% per week) stage and the second stage (about 1 wt% per week), due to the faster degradation rate of CS. This scaffold was implanted into a femur bone defect of a rat. It degraded after 13 weeks of implantation, and new bone and angiogenesis were observed [81]. Organic materials such as collagen are more easily degraded than inorganic compounds such as bio-active glass and nHA. Thus, these two factors are often combined to obtain satisfactory degradation rates. In addition to organic materials, growth factors such as recombinant human bone morphogenetic protein (rhBMP2) have also been reported to promote degradation [82]. The factors influencing the degradation of scaffolds are listed in Table 2. To observe the process of HA degradation, Li et al. prepared a concentric membrane with an internal HA/terbium (TB) ring and an external ring of HA/nano-wires (NWS). The interior and exterior ring materials are fluorescent at the same time. By tracing the fluorescence distribution, HA/TB initially diffused from the internal ring, indicating that HA/TB degraded earlier than HA/NWS [83]. This provides a reference for understanding a series of behaviors such as degradation, diffusion, and reconstruction of a scaffold during bone repair. There have been relevant studies on in vitro/in vivo degradation of scaffolds, but the mechanisms related to cell degradation of scaffolds are not clear. Li et al. [84] prepared nHA/COL composite scaffolds and loaded umbilical cord MSCs. In this study, the inhibition of the Wnt/β-catenin pathway was found to lead to rapid degradation of scaffolds and delayed new bone formation. This provided insight for studying the biological degradation mechanisms of a scaffold. Extensive studies have shown that it takes about two to four months to observe new bone formation and takes up to six months to obtain histological healing [25,85,86,87]. Therefore, the degradation rate of scaffold materials should correspond to this rate. The degradation of materials can be adjusted by preparing cross-linked composite scaffolds and can also improve the elastic modulus of the scaffolds [88,89,90,91]. In addition, HA can be doped with metal dopants to improve their properties; for example, Mg^2+^ prevents HA crystallization and reduces bone brittleness. Zn^2+^ can inhibit bacterial colonization and regulate inflammatory mediators. Another metal dopant, Sr^2+^, promotes bone formation by activating signaling pathways, inhibiting bone resorption and regulating cell behavior [92,93].

### 2.4. HA Scaffolds Combined with Cells/Extra-Cellular Matrix/Cytokine to Promote Osteogenesis

Seed cells and cytokines are major components of bone tissue engineering. We mentioned that the porosity of the scaffold is associated with the entrance of cells and growth factors. Some scholars directly combined cytokines, cells, or extra-cellular matrix with HA, hoping to obtain better biocompatibility and osteogenic properties.

BMSCs are pluripotent stem cells that can differentiate in the direction of osteogenesis, lipolysis, and cartilage, and their ability for multi-directional differentiation also makes BMSCs the most commonly used seed cells. Wüst et al. encapsulated hBMSCs using HA/alginate/gelatin composite hydrogels and used 3D printed gel scaffolds. The scaffolds were then cultured in vitro for three days, and cellular viability was identified by fluorescent staining. Most of the cells in the material were alive, and only 15% were dead cells [107]. In addition to bone marrow, mesenchymal stem cells derived from the umbilical cord and adipose tissue were applied [4,108]. Osteoblast lines or osteoblastic progenitor cells lines such as MC3T3-E1 and MG-63 are often used in tissue-engineered bone [109,110]. Bone homeostasis is maintained by the balance of osteoblasts and osteoclasts, and current research focuses on the osteoblasts. However, previous studies have shown that osteoclasts not only absorb bone, but also regulate osteoblasts to promote osteogenesis [111]. Thus, further development of tissue-engineered bone may introduce osteoclasts to promote bone remodeling.

In tissue engineered bone, seeding of cells onto scaffold material and then implanting the cell/scaffold complex into the defect area is a common method. Some scholars have sown MG-63 osteoblasts on the surfaces of titanium, stainless steel, and collagen-coated materials with different roughness. It was found that cells on the collagen coating had more local adhesion, and the moving speed could reach 60 nm/min [110]. The increase in focal adhesions suggested that the cells have strong dynamic capacity and ECM secretion capacity. This study also suggests that an extracellular matrix may also influence the migration and osteogenesis of seed cells. Onishi et al. [112] used an osteoblast sheet to generate extracellular matrix plates that bind cytokines such as BMP-2 and transforming growth factor β1 (TGFβ1), and it was implanted into the rat femur defect site. The study found that the osteoblast extracellular matrix loaded with growth factors such as BMP2 and TGFβ1 first appeared at bone deposition, and the maximum bending load at eight weeks was significantly stronger than that of the control group, which is a rare scaffold-free extracellular matrix to repair bone defects. Over the past few years, self-assembled molecular materials have been able to effectively support the proliferation and differentiation of various cells. Self-assembling peptides are nanostructured and biomechanically similar to extracellular matrices, and can form a 3D environment for cells. This makes them suitable candidates for artificial cellular niches, which have been applied in bone tissue engineering [113,114]. Alshehri et al. mixed hBMSCs with IVZK-IVFK peptide solutions to form 3D cultures. This self-assembling peptide was found to promote osteogenesis differentiation of BMSCs [114]. Zhang et al. [115] combined self-assembled peptides with nHA/CS scaffolds, which increased BMSCs adhesion and improved the mechanical properties of the scaffolds. It was implanted into the femoral defect of a mouse, and the defect healed after 12 weeks. A large amount of trabecular bone was found in the newly formed bone. It is suggested that self-assembled peptides may promote osteogenesis and have good prospects in tissue-engineered bone.

BMP, a cytokine that induces bone formation, was extracted from human demineralized bone matrix by Urist [116]. It can regulate cell proliferation, differentiation, apoptosis, or morphogenesis and possesses various biological functions. BMP2 belongs to the first sub-type and is primarily involved in bone growth and differentiation. Many scholars combine it with scaffolds to promote the induction and differentiation of BMSCs. Su et al. [82] developed collagen/HA nano-composites and pre-absorbed rhBMP2, implanted them into the femoral defect of rabbits, and collected samples for analysis after eight weeks. Micro-CT and histological staining indicated that the scaffolds with pre-absorbed rhBMP2 could effectively repair bone defects. Bone formation occurred faster and it reached a higher degree of mineralization than the rhBMP2-free material. Composite scaffolds made of BMP2 and other materials have the capacity to promote adherence proliferation and osteogenic differentiation of cells, and they have good osteogenic induction properties [117,118,119,120,121,122]. Some scholars have also synthesized BMP-2 peptidomimetics (E7 BMP-2 peptides) [123], p24 peptides derived from BMP2 [124] and p28 peptides [109], which can also promote cell proliferation, adherence and differentiation, and promote bone tissue formation in a dose-dependent manner. Shui et al. [125] introduced recombinant adenovirus into osteogenic progenitor cells to mediate the expression of BMP9. The BMP9-mediated C2C12 cells were found to easily develop mineralized nodules. BMP9 can indeed help differentiate C2C12 cells into osteoblasts. Meanwhile, collagen sponges loaded with BMP9-transduced C2C12 induced strong bone formation within four weeks. Apart from the BMP family, there are many other cytokines that promote osteogenesis. Liang et al. [126] used naringenin (NG) and calcitonin gene-related peptide (CGRP) in combination with HA and sodium alginate (SAL) composite hydrogel scaffolds and demonstrated that they have better biosynthesis than simple composite scaffolds in bone function. MicroCT, HE and TB staining showed a significant increase in the amount of mineralization infused with NG and CGRP. Shi et al. [127] functionalized HA with lactoferrin (LF), which could enhance cell viability and have high biocompatibility. Typical growth factors used in tissue engineering are presented in Table 3 [34,82,117,118,119,120,121,122,125,126,128,129,130,131,132,133,134,135,136,137,138,139,140,141,142,143,144,145,146,147,148,149,150,151,152,153,154]. More importantly, the incorporation approaches and release profiles of growth factors govern the differentiation of cells and are closely correlated with osteogenesis.

The function of cytokines has been basically clarified, and they have been extensively used in vitro and in vivo. However, the interaction mechanism and the spatial-temporal release of multiple cytokines require further exploration. Currently, the major part of the assessment of osteogenesis effects adopts histopathologic indicators given a lack of more accurate and quantitative methods, and research on the mechanism of osteogenesis is insufficient. These factors limit the clinical application of tissue-engineered bone.

## 3. HA Compound Bioscaffolds Promote Angiogenesis

Based on bone regeneration, scholars have further investigated the vascularization of tissue-engineered bone. Regeneration of vessels can effectively transport nutrients and metabolic waste, which is of great importance in the formation of bones. Currently, there exist three primary strategies to promote vascularization: (1) seeding angiogenic cells such as endothelial cells; (2) adding cytokines that guide angiogenesis, such as vascular endothelial growth factor (VEGF), etc.; (3) through microsurgical techniques, transplanting autologous vessels to promote vascularization. The above three methods have received great attention by scholars in the vascularization of tissue-engineered bone and have been extensively utilized. The strategies of vascularization and neuralization of tissue-engineered bone are shown in Figure 4.

### 3.1. HA Scaffold Combined with Vascular Endothelial Cells Promotes Angiogenesis

With further research on tissue-engineered bone vascularization, scholars co-cultured angioblasts and osteoblasts and formed complex scaffolds. These experiments were extensively done and gave good results [155]. Zhang et al. [156] developed a double-cell sheet (DCS) complex consisting of osteoblast sheets and vascular endothelial cell sheets with osteogenesis and vascular formation potential. The double-layer diaphragm and coral hydroxyapatite (CHA) complex were implanted under the skin of nude mice, and it was found to have more mature mineralization and generated more capillaries after 12 weeks. The success of in vivo vascularization has brought tissue-engineered bone closer to clinical application. Vascular endothelial cells and osteoblasts participate in bone tissue engineering together, which can promote angiogenesis and improve the quality of bone [157].

The use of endothelial cells and osteoblasts, co-cultured to promote vascularization and osteogenesis, has been widely explored. Histologically, a lumen-like structure was seen, which indicated some progress in a positive direction. However, the vessels are still relatively naive, and it remains to be determined if the newly formed vessels have blood perfusion and nutrient exchange. Furthermore, some scholars have found that nHA can accumulate in vascular endothelial cells, thereby reducing the ability of cells to form blood vessels, which is a problem that requires attention [158].

### 3.2. HA Scaffolds Combined with Pro-Angiogenic Growth Factors or Drugs to Promote Angiogenesis

It is common knowledge that secreting pro-angiogenic growth factors may induce angiogenesis. Among these, the common cytokines are: VEGF [141], fibroblast growth factor (FGF) [1], angiopoietin (Ang) [143], transforming growth factor (TGF) [148,159], and platelet-derived growth factor (PDGF) [160], among others. In order to induce vascularization, scholars have compounded VEGF on the scaffold or transfected the VEGF gene into BMMSCs to promote the secretion or sustained release of angiogenic factors [161,162]. Quinlan et al. [163] compounded VEGF-containing alginate microparticles (MPs) with COL/HA scaffolds. Long-term stable release of VEGF occurs for up to 35 days. The composite scaffold was implanted into the rat bone defect model. Eight weeks later, obvious angiogenesis was subjected to immunofluorescence staining. To ensure the spatial-temporal release of angiogenic growth factors, the gene encoding cytokines were transfected into seed cells. For example, BMSCs were transfected with lentiviral vectors (LV-pdgfb) and seeded on PLGA/nHA scaffolds. It was found that the expression of pdgfb and the angiogenesis-related genes vWF and VEGFR2 was significantly increased, and the formation of new bone and vessels was significantly increased in vivo [144]. Some drugs such as salvianolic acid B [164] can promote the release of VEGF and induce angiogenesis.

Abundant experimental evidence exists for the use of growth factors in promoting angiogenesis. However, there is still no gold standard for selecting cytokines, and different cytokines may promote angiogenesis at different degrees for different cells. Combined cytokine application is better than single application, and the specific mechanism of action is still unclear. Moreover, the regulation of vascular growth by angiogenic growth factors or cells remains an urgent issue to be resolved in tissue-engineered bone vascularization.

### 3.3. Microsurgical Techniques to Promote Angiogenesis

Revascularization through microsurgical techniques is also a method of vascularizing tissue-engineered bone. The most common techniques are the implantation of the vascular pedicle, the implantation of the arteriovenous ring, the implantation of the periosteal flap, the implantation of the muscular flap, and packing of the fascial flap [165,166]. 

Due to the presence of the blood supply outside the tissue-engineered bone, the regeneration of the internal blood vessels and the formation of the bones are promoted. Largo et al. [167] co-cultured vegf-transfecting rBMSCs with HA and placed them in the muscle of rabbits. After one week, MRI examination revealed increased tissue perfusion and generation of a dense capillary network in the graft. Vascular bone muscle flaps with vascular peduncles can be obtained. Animal experiments also showed that the bone volume produced by the periosteum is greater than that of the muscle. Tissue engineered bone implanted in the periosteum can form new bones without exogenous growth factors. Consequently, some scholars have implanted tissue-engineered bone into the periosteum. Following the formation of new bone, the supplying arteries and veins were harvested together to obtain a vascularized bone flap [168]. Researchers have made tissue-engineered bone from porous HA compound bone marrow aspiration fluid and BMP2. It is wrapped in the periosteum and has been implanted in the large omentum of small pigs. Bone and angiogenesis can be seen at 8 to 16 weeks postoperatively, suggesting that the use of periosteum can promote bone formation with grafting. The large omentum is a bioreactor suitable for in vivo culture of tissue-engineered bone. Because of the large vascular supply, the resulting bone flap can replace autologous bone flap grafting [169]. Tatara et al. [170] mixed commercialized materials (15% HA and 85% β-TCP) with autologous bone in different proportions. The composite stent is first placed at the site of the rib defect in the sheep and in contact with the rib periosteum to promote formation of new bone. After nine weeks of implantation, tissue-engineered bone with intercostal arteries and venous vascular pedicles was obtained and implanted into the site of the mandible defect. After microvascular anastomosis and fixation, 3/3 of the animals survived after 12 weeks. This cell-free, factor-free strategy sheds light on the reconstruction of bone defects. However, the sample size of the study is small, and the graft bone did not play the role of load-bearing bone, so further in-depth research is needed. Winkler et al. [171] anastomosed naked mice cryptovenous arteriovenous, making it an arteriovenous vessel ring. Human adipose-derived stem cells (hADSCs), human umbilical vein endothelial cells (hUVECs), and HA after osteogenesis induction were implanted into the vascular ring. Significant vascular formation was observed. Creation of arteriovenous rings (AVs) and deep fascial flaps with vascular pedicles avoids the harvesting of the flap and has been used in large animal models [172] and humans [173]. Even so, it takes several months for the bone to become vascularized, which limits its clinical application.

Although numerous experimental studies have confirmed that the various methods mentioned above can promote the vascularization and osteogenesis of tissue-engineered bone, methods such as vascular bundle implantation, prefabricated vascularized muscle-skeletal flaps, and the use of fascial flaps or muscle flaps to wrap artificial bone not only require higher microsurgical techniques, but also have unclear indications. There are still many limitations to its clinical application.

## 4. Construction of Innervated Tissue Engineered Bone

Domestic and foreign studies have found that nerve growth is parallel to bone growth and development [174]. Nerves are also closely associated with mature bones. There exists a rich neural network in the periosteum that is distributed in the Haversian system [175]. The nerves are distributed in some areas of active bone metabolism, such as the bone marrow, periosteum, and epiphysis. Coupaud et al. [176] found that the bone mineral density of the tibia, femoral cortex, and trabecular bone decreased significantly after a one-year follow-up in patients with spinal cord injury. These studies suggest that nerves have a significant influence on bone development and metabolism.

Nerve fibers of the innervating bone are divided into sympathetic and sensory nerves, which regulate the bone environment and osteocytes through neurotransmitters [177]. Our previous studies confirmed that sympathetic nerves play a negative regulatory role. Tensile stress promotes the detachment of local BMSCs from sympathetic/endothelial cell-based stem cell niches and migration to the osteogenic front through chemokines such as SDF-1 [178]. Sensory neurons play a key role in regulating bone homeostasis. Their injury results in decreased new bone formation and enhanced bone resorption, ultimately leading to increased bone fragility [179]. Nerve growth factor (NGF) can promote mandibular sensory nerve recovery and bone regeneration during distraction osteogenesis [180]. Sensory nerves play an important role in bone regeneration and reconstruction. Sensory nerves regulate bone homeostasis through secreting neurotransmitters, which include NGF, calcitonin gene-related peptide (CGRP), substance P (SP), and semaphorin 3A (Sema3A) [181]. These neurotransmitters bind to receptors on the cell membrane of mesenchymal stem cells to regulate cell migration to the osteogenic front and promote osteogenesis.

When an autologous bone graft is used to reconstruct jaw defects, severe osteoporosis occurs in the grafted bone. Even if vascular anastomosis is performed to ensure sufficient blood supply, osteoporosis still happens. This phenomenon indicates that the internal environment of the grafted bone may be controlled by systemic factors other than blood supply. In our previous study, we developed an innervated bone graft by dissecting and anastomosing the nerves of the donor and recipient. Retrospective and prospective clinical studies have shown that simultaneous innervated bone grafting can significantly reduce osteoporosis and effectively maintain lower lip sensation. It also suggests that nerves are important in bone homeostasis [182,183]. In recent years, studies in the field of tissue engineering have found that in addition to vascularization, neurogenesis may also be important in promoting the formation of bone [167,184,185].

Neural stem cells (NSC) and NGFs were loaded on silk-HA composite scaffolds. After three days, the gene expressions of NSC marker Nestin and differentiated neuron markers Tubulin β3 (Tubb3) and microtubule-associated protein 2 (MAP2) were observed by qPCR (quantitative polymerase chain reaction). There was no statistical difference in Nestin’s expression when compared with the untreated nerve growth factor group, but Tubb3 and MAP2 were significantly higher than the control group. NGF is proposed to be an important factor in neurogenesis [186]. Coincidently, some researchers used CS/HA scaffolds loaded with NGF in the mandibular canine–molar region. NGF-loaded scaffolds were found to regulate gene levels related to NGF, osteogenic differentiation, and neurogenic differentiation. NGF significantly induces osseointegration and neuronal regeneration [187]. Liu et al. [188] designed a HA membrane enriched with nanocellulose to induce the differentiation of MSCs into osteoblasts and neuronal cells in vitro. Bioactive tissue-engineered bone containing osteoblasts, neuronal cells, and endothelial cells was also simulated in vitro. And through heterotropic osteogenesis, the rapid formation of blood vessels, nerve fibers, and new bone was found. There was evidence that the membrane can be used to induce the differentiation of MSCs, which has a good prospective application in bone tissue engineering. Cui et al. [189] implanted tissue-engineered bone with a vascular bundle and sensory nerve tract in two separate groups, and simple tissue-engineered bone was used as control. Following the implantation into the femoral defect of rabbits, the expressions of calcitonin gene related peptide (CGRP) and neuropeptide Y (NPY) were observed by immunohistochemical staining. The expression of CGRP and NPY in the tissue-engineered bone implanted with vascular bundles alone was higher than in the nerve bundle implantation group at three months. At 6 to 12 months, there was no significant difference in their expressions between the two groups, and they were all higher than those in the control group. Some scholars have also implanted sensory nerve bundles and motor nerve bundles into tissue-engineered bone to repair femoral defects in rabbits. The bone mineral density (BMD) test at 12 weeks after surgery showed that the bone defect in the sensory nerve tract implantation group was well repaired, and the BMD value was higher than that in the motor nerve tract implantation group and tissue-engineered bone control group [190]. In addition, the research team also found that Schwann cells can specifically promote BMSCs-derived endothelial cells to generate blood vessels, providing a cytological basis for neural tissue-engineered bone construction [191]. Implanted sensory nerve fibers can quickly grow into the scaffolds, increasing the expression of neurotransmitters such as CGRP [185]. Implantation of vascular bundles or sensory nerve bundles can significantly enhance the vascularization and neurotization of tissue-engineered bone for better osteogenesis [175,184].

The importance of sensory nerves in bone physiology and the regulation of the bone repair process indicate the correlation between sensory nerves and bone. Nerve tissue-engineered bone has good potential for application in the development of large-scale bone repair. Since the nerve distribution and mechanism of action in normal bone tissue are not yet clear, further research in basic medicine is required. In addition, the research on neutralizing tissue-engineered bone is still in the animal experiment stage, and its clinical application needs to be further explored.

## 5. HA Composite Commercial Products

Collagraft^®^ is a clinical alternative to autologous bone grafting. It is primarily a mixture of HA and β-TCP (13:7, mass ratio) with the addition of highly purified Col-I and prepared in a sterile lyophilized solution. In sheep lumbar models, Collagraft^®^ with or without bone marrow produced new bone with mechanical properties similar to those of autologous bone grafting and with higher BMD [192]. When combined with syvium-derived stem cells (SDSCs), fibrin glue, and poly (glycolic acid) (PGA), Collagraft^®^ can also be used to repair cartilage defects [193]. When combined with antibiotics, Collagraft^®^ maintains 73% vancomycin activity and 61% gentamicin activity, and releases 432μg/mL vancomycin or 301μg/mL gentamicin within 48 h [194], making it suitable for topical delivery of antibiotics to prevent infection. In addition, compared with OsteoSet^®^ (calcium sulfate), ProOsteon^®^ (ceramics), and DBX^®^ (demineralized bone matrix), Collagraft^®^ has a stronger ability to induce bone remodeling and bone tissue regeneration [195,196]. Especially when combined with bone marrow, osteogenesis is more obvious [197]. A prospective randomized controlled clinical study of 267 patients with long bone fractures found that Collagraft^®^ showed the same efficacy as autogenous iliac graft in the treatment of acute long bone fractures over 6–12 months [198]. However, other studies have shown that Collagraft^®^ is not effective at inducing osteogenesis. Osteoid formation was found in 80% (4/5) of HA-TCP scaffolds with adipose-derived adult stem cells (ADAS) compared to 20% (1/5) of Collagraft^®^ loaded with ADAS. 100% (3/3) of the HA-TCP implants loaded with hFOB 1.19 cells formed osteoid, compared to only 1/3 in Collagraft^®^, and significant adipose tissue was found in the Collagraft^®^ [199]. Collagraft^®^, approved for the treatment of acute long bone fractures, healed significantly less than autogenous cancellous bone in a spinal fusion model, which suggests that clinicians should carefully consider using Collagraft^®^ for spinal fusion [200].

Ossceram^®^ Nano is a safe, absorbable bone replacement material composed of HA and β-TCP (3:2, mass ratio). β-TCP was rapidly decomposed while HA was not easily degraded. As a result, Ossceram^®^ Nano has more ions released into the environment at the early stage of implantation compared to HA alone, allowing more collagen fiber colonization and promoting new bone maturation [201]. In addition, Ossceram^®^ Nano has a microporous structure that facilitates ion exchange, a macroporous structure that facilitates cell colonization and vascular growth, and a nanostructure that supports bone formation [201], making it an effective bone substitute material.

Geistlich Bio-Oss^®^ is a high-purity HA-based biological material extracted from natural bovine bone, which is widely used for alveolar bone defects, increasing bone mass before dental implants, and repairing craniofacial bone defects [202,203,204,205]. Bio-Oss^®^ exhibits an inorganic composition and porous structure similar to human cancellous bone, facilitating the formation and growth of new bone.

Several other commercial HA composites are listed in Table 4. Although some clinical studies have reported the efficacy of commercialized HA composites as an alternative product for bone grafting, most of these studies have been observational, and high-quality randomized controlled studies are still needed.

## 6. Conclusions

Over the past few years, bone tissue engineering technology has expanded rapidly. HA occupies an irreplaceable position in scaffolds due to its good mechanical strength and biocompatibility. As mentioned above, in order to achieve a better osteogenic effect, HA composite scaffolds can be developed to be nearly similar to the composition and mechanical strength of natural bone. The ideal compressive strength is 2–230 MPa, and the ideal elastic modulus is 0.05–30 GPa for HA composites, which matches with the natural bone. The surface morphology and pore diameter that best suit the differentiation of seed cells can be explored. In general, pore size above 50 μm facilitates the entry of cells and angiogenesis. Additionally, the rate of material degradation can be adjusted to match the rate of bone regeneration, which takes about 2–6 months. However, there are numerous fields that need further study: 1. Compared with the mechanical strength of the scaffold before implantation, dynamic changes of the strength after implantation deserve attention; 2. Optimizing the creation process and low temperature disinfection may help to prepare a scaffold with both a porous structure and excellent mechanical properties; 3. HA can be doped with metal dopants such as Mg^2+^, Zn^2+^, Sr^2+^, etc., to improve their properties; 4. The interaction mechanism and the spatial-temporal release of multiple cytokines require further exploration. Hence, other demands for function make biomimetic tissue-engineered bone a hot topic. Biomimetic tissue-engineered bone is gradually approaching the qualities of physiological bone, and some progress has been made in angiogenesis and neurogenesis. Regenerating blood vessels and nerves can not only serve as a guarantor of bone formation, but also provide tissue-engineered bone with sensory functions. There is no doubt that breakthroughs in vascularization and innervation will enhance the vitality and prospects of tissue-engineered bone.

## Figures and Tables

**Figure 1 materials-15-08475-f001:**
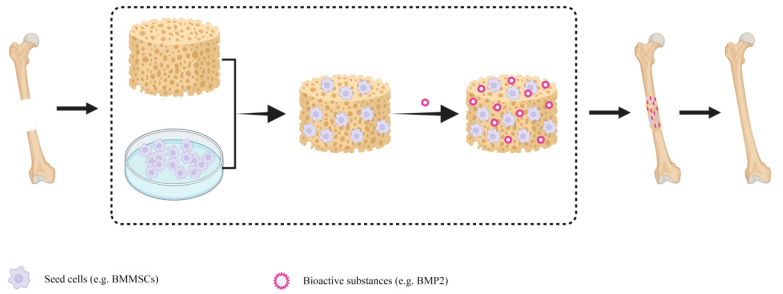
Schematic diagram of tissue engineering bone composition and repair of bone defect. The cells were cultured on scaffolds, and then some bioactive substances such as cytokines were added. The tissue-engineered bone was implanted into the bone defect, and then the new bone was generated by degradation of the scaffold material, and the bone defect was repaired. BMMSCs-Bone marrow mesenchymal stem cells; BMP2-Bone Morphogenetic Protein 2.

**Figure 2 materials-15-08475-f002:**
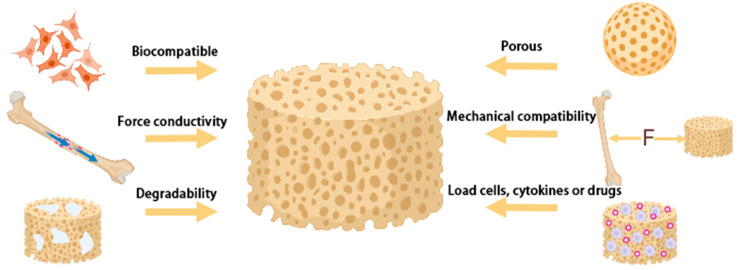
Requirements for bone tissue engineering scaffold materials. Scaffold materials should be biocompatible, that is, non-toxic and harmless to cells; they should force conductivity; be degradable; be porous, used to contain cells and cytokines, and can also be used for the discharge of nutrients and metabolic waste; have mechanical compatibility, that is, having similar mechanical strength and elastic modulus with natural bone; and have the ability to load seed cells, cytokines or drugs.

**Figure 3 materials-15-08475-f003:**
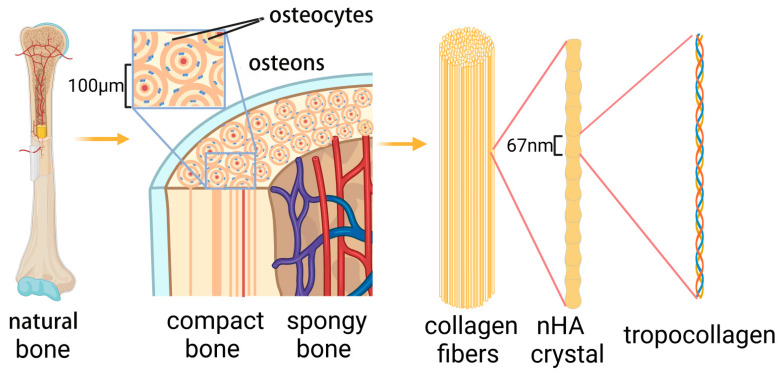
Hierarchical structure of bone from macroscale to molecular composition. Natural bone is primarily composed of compact bone and spongy bone. Compact bone includes external circumferential lamellae, internal circumferential lamellae, osteons (Haversian canals) and interstitial lamellae. Osteocytes are surrounded by osteons. Bone has a lamellar structure, and a single lamella is composed of collagen fibers; three amino acid chains and nHA form collagen molecules (procollagen).

**Figure 4 materials-15-08475-f004:**
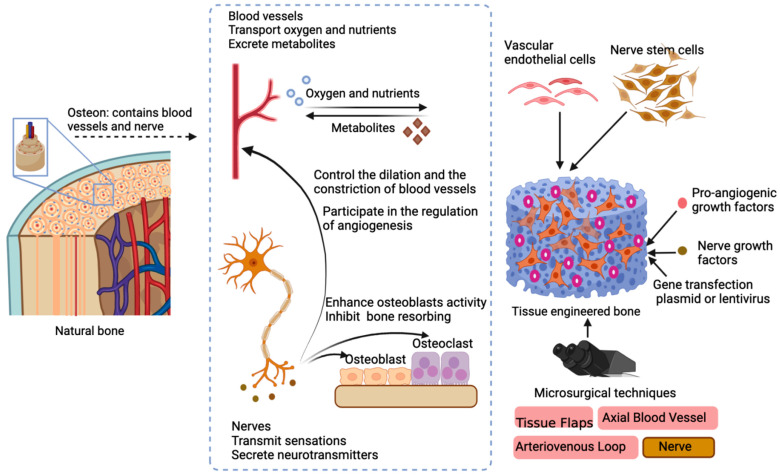
Vascularization and neuralization of tissue-engineered bone. Natural bone contains blood vessels and nerves. Blood vessels can transport oxygen, nutrients, and metabolites, which is very important for the proliferation and differentiation of seed cells in tissue-engineered bone. Nerves can transmit sensations and secrete neurotransmitters. Tissue-engineered bone can possess sensation, which makes it more bionic. Neurotransmitters can regulate the activity of osteoblasts and osteoclasts to maintain bone homeostasis. It can also regulate the constriction and relaxation of blood vessels, which are mainly determined by the type of blood vessels. Neurotransmitters are involved in regulating the regeneration of blood vessels. The current strategies for vascularization and neuralization of tissue-engineered bones mainly include: loading vascular endothelial cells and neural stem cells; loading angiogenesis factors and nerve growth factors, which can also be transfected into cells by transfecting gene coding; implanting blood vessels, nerves, and other tissues through microsurgery.

**Table 1 materials-15-08475-t001:** Mechanical properties of HA composites.

Material	Ratio	Porosity and Pore Size	Compressive Strength (MPa)	Flexural, Tensile Strength (MPa)	Fracture Toughness (MJ/m^3^)	Elasticity Module (GPa)	Ref.
Cortical bone	-	Porosity: 5–10% Pore size: 10–50 μm	100–230	50–150	2–12	7–30	[35,37,38]
Cancellous bone	-	Porosity: 75–85% Pore size: 300–600 μm	2–12	10–20	-	0.5–0.05	[35,36]
HA	-	Porosity: 63.3–78.4%	95	161	1.1	-	[39]
PLA/HA	8:2 (*w*:*w*)	Porosity: 83.0%	34.1	-	-	2.63	[40]
HA/PAAm/Dex	-	-	6.5	-	-	-	[23]
HA/β-TCP	1:1 (*w*:*w*)	Porosity: 84.9% Pore size: 698 μm	1.7	-	-	-	[41]
nHA/CS	1:50 (*w*:*v*)	Pore size < 10 μm (Lyophilized) Pore size < 2 μm (Lyophilized Soaked and Dried)	3.93 (Lyophilized) 8.59 (Lyophilized Soaked and Dried)	-	-	-	[42]
nHA/alginate/CS	70:18:12 (*w*:*w*)	Porosity: 81.0% Pore size: 100–400 μm	4.42	-	-	13.35	[43]
PEEK/HA	8:2 (*w*:*w*) 6:4 (*w*:*w*)	Porosity; 60.4–87.8% Pore size: 200–2000 μm	2.2–35.2	-	-	0.05–0.62	[44]
PEEK/HA	79:1 (*v*:*v*)	-	-	170 (non-linked composites) 171.7 (covalently linked composites)	-	4.8 (non-linked composites) 5.0 (covalently linked composites)	[45]
Silane/HA/Gel	-	-	97 195 (THF) as the cosolvent)	222 431 (THF) as the cosolvent)	-	-	[46]
Aac/GO/HA/TiO_2_	-	Porosity: 79.97–44.32% Pore size: 107.42–256.11 μm	2.96–13.31	-	-	0.04–0.3	[47]
nHA/CS-TSP	7:2:1 (*w*:*w*)	-	4.0	-	-	-	[48]
COL-HA-MFC	-	Porosity: 79.96–87.55% Pore sizes: 50–100 μm	20–40	-	-	-	[49]
EC/nHA	nHA concentrations: 3, 5, and 10% (*w*/*w*)	-	-	70.6	-	3.12	[50]
PVA/CS/CHA	CHA concentrations: 0, 5, 10, 15, 20% (*w*/*w*)	-	7.49	-	-	0.10	[51]
HA/PLGA	9:11 (*w*:*w*)	Porosity:60% Pore sizes: 359.4 μm	40	-	-	-	[52]
Gellan/guar gum/HA	2:1:5.44 (*w*:*w*)	-	12.662	-	-	-	[53]
HA/PLGA/PGA	1:9:23.33 (*w*:*w*)	Pore sizes; 3 μm	31.1	-	-	-	[54]
Silk/HA	HA concentrations: 80, 90, 99% (*w*/*w*)	Porosities; 62.9%	8.7–152.4	-	-	-	[55]
PEGDA/HA	HA concentrations: 0, 0.5, 1, 1.5, 2% (*w*/*w*)	-	6.5	-	-	-	[56]
HA/GCPU	2:3 (*w*:*w*)	Pore sizes: 500 μm	4.6	-	-	-	[57]
nHA/CS	-	Pore sizes: 0–80 μm	9.41	-	-	0.17	[58]
Cellulose-graft-polyacrylamide/nHA	-	Porosity: 47.37% Pore sizes:120–190 μm	4.80	-	-	0.29	[59]
SA/HEC/HA	40 wt%HA	Porosity: 66.7% Pore sizes: 100–300 μm	23.9	-	-	-	[60]
TiO_2_/grafted cellulose/nHA	-	Porosity: 80–87% Pore sizes: 70–80 μm	4.1	-	-	1.20	[61]
Mgo/nHA/PLLA	-	-	-	-	-	1.00	[62]
nHA/CS/TFSP	-	Porosity: 60.3%	6.7	-	-	-	[63]
HA/BT	1:9 (*v*:*v*)	Porosity: 57.4%	14.5	-	-	-	[64]

PAAm-polyacrylamide; Dex-dextran; PEEK-Polyetheretherketone; Aac-Acrylic acid; GO-graphene oxide; TSP-tamarind seed polysaccharide; MFC-Microfibrillated cellulose; EC-electrospun cellulose; PVA-poly(vinyl alcohol); CHA-carbonated hydroxyapatite; PLGA-poly(lactic-co-glycolic acid); PGA-poly(glycolide); PEGDA-poly(ethyleneglycol) diacrylate; SA-Sodium alginate; HEC-hydroxyethylcellulose; PLLA-poly(l-lactic acid); TFSP-*Trigonella foenum graecum* seed polysaccharide; BT-barium titanate; Ref.-Reference.

**Table 2 materials-15-08475-t002:** Factors influencing the degradation of scaffolds.

Scaffolds	Parameters	Results	Ref.
nHA/CS/Gel	Polymer ratio	Among the different proportion of the nHA and CS/Gel, HA80CG20 (nHA:CS/Gel = 8:2) possess the highest degradation.	[94]
PHB/nHA	Shape	The smaller the thickness of the specimen, the higher the aspect ratio and therefore the greater surface area for the enzymes to attack.	[95]
PLA/HA	Surface area of particles	15% PLA scaffolds filled with HA (spray dried) exhibited significantly lower mass loss rate than scaffolds filled with HA (sintered), which was possibly due to the higher surface area of particles.	[96]
CS/1wt%HA(CAP)	Porosity/pore size	Larger surface area and more active sites for the lysozyme reaction were obtained through CAP treatment, which increased the degradation rate significantly.	[75]
PGA/HA/PLLA	Hydrophilic	The PGA enhanced the degradation rate of scaffolds as indicated by increasing the weight loss, owing to the degradation of high hydrophilic PGA.	[79]
CS-g-PCL		Hydrophilic materials including copolymer and nHA were compounded with PCL to increase the degradation rate. Compared with pure PCL, PCL90:n5 showed 10% weight loss.	[97]
PLGA/HA/βTCP		PLGA copolymers are less hydrophilic and lead to a slower degradation of the polymer chains.	[98]
PLA/HA	PH	Alkaline HA particles release OH− ions, which can neutralize the acidic degradable substances generated during the PLA degradation process. As a result, acid driven autocatalytic degradation of the polymer phase can be reduced, thus slowing degradation.	[96,99]
nHA/PCL/Pluronic		The nHA/PCL/Pluronic scaffold had the lowest degradation rate in pH 7.4 among three degradation media. The solution of HA particles in acidic environment accelerated the degradation of the polymer matrix just as the self-catalysis of acidic products on polyester degradation. At high pH, PCL has further advantages as it introduces OH and COOH groups on the surface, rendering hydrophilic substrates that will prove its degradation.	[100]
Sr10-HA- g-PBLG	Relative molecular weight	The molecular weight of the grafted PBLG increased with the theoretical grafting ratio. The degradation rate decreased with the increasing molecular weight of grafted PBLG.	[101]
ARX/BG/AAc/GO/nHA	Physical crosslinked	nHA increased physical crosslinkers among the polymeric materials and had the reverse impact on degradation. The scaffold sample BGH1 (contain 1.4 g nHA) showed the most degradation, while BGH3 (contain 1.6 g nHA) showed the least.	[102]
nHA/alginate/GEL		Alginate strengthened the resultant hydrogel stability by increasing crosslinking densities, thus reducing the degradation rate.	[103]
nHA/CS/Gel	Interactions between materials	In the presence of CA, the weight loss of the scaffolds decreased, probably due to the increased interaction between nHA particles and CS and Gel molecules via CA functional groups.	[94]
PCL/PHBV/nHA		PCL70/PHBV30/nHA5 (wt%) maintained higher degradation than PCL90/PHBV10/nHA5 (wt%), which indicated the weak interactions between two polymers.	[104]
GP/nHA/HPCS	Covalent linkage	GP can increase the covalent linkage between polymer chains, thereby reducing the degradation rates.	[91]
GelMA/PEGDA/nHA		Ca^2+^ in nHA can coordinate with the amide bond of gelatin, thereby increasing the stability of the composite hydrogel and prolonging the degradation time.	[105]
HA/rGO	Biocompatibility	Good biocompatibility provided by the rGO can improve cell adhesion and promote the proliferation and differentiation of cells. The new bone formation permits the extrusion of new bone, which supports cell adhesion and osteogenesis.	[73]
Si/nHA	Biological effect	Composite matrix with Si/nHA presented superior biodegradation of the scaffold. Silicate species released from the Si/nHA scaffold would attract osteoclast cells toward the biomaterial.	[106]
CS/HA	Growth factors loading	RhBMP-2 and concomitant rapid material degradation synergistically promote bone repair and regeneration with CS/HA scaffolds.	[82]

PHB-poly(3-hydroxybutyrate); PLA-poly(lactide acid); CAP-cold atmospheric plasma technology; PBLG-Poly(γ-benzyl-l-glutamate); ARX-arabinoxylan; BG-β-glucan; CA-citric acid; PHBV-poly (hydroxybutyrate-co-hydroxyvalerate); GP-genipin; HPCS-Hydroxypropyl chitosan; GelMA-gelatin methacrylamine; Ref.-Reference.

**Table 3 materials-15-08475-t003:** Growth factors commonly used in tissue engineering.

Growth Factor	Scaffolds	Incorporation Approach	Release Profile	Outcomes	Ref.
BMP-2	CS/HA	Physical entrapment/Adsorption.	Rapid burst release between 12–36 h, followed by a slower release during the following 7 days.	In vitro: RhBMP-2 and concomitant rapid material degradation synergistically promote bone repair and regeneration with CS/HA scaffolds.	[82]
BMP-2	CS/HA/TiO2	Physical entrapment/Adsorption.	Rapid burst release in the first week, followed by a slower release during the following 3 weeks.	In vitro: The BMP-2 loading in CS can significantly improve cell adhesion, spreading, and proliferation.	[117]
BMP-2	HA/PLL/PDA	Physical entrapment/Adsorption.	Rapid burst release in the first week, followed by a slower release during the following 2 weeks.	In vitro: The BMP2-entrapped PLL/PDA coating on the HA scaffold can promote osteogenic differentiation of BMSCs. In vivo: Induce ectopic bone formation to a much greater level in vivo compared with a bare HA scaffold that delivers BMP2 in a burst manner.	[118]
BMP-2	nHA/COL	Physical entrapment/Adsorption.	Extended sustainable release for 21 days.	In vitro: The BMP-2-nHA-COL scaffold promoted BMSCs adhesion, proliferation, and differentiation.	[119]
BMP-2	PCL/PDA/HA	Physical entrapment/Adsorption.	Approximately 20% of the BMP-2 was rapidly released within 24 h.	In vitro: Increased osteoblast proliferation and osteogenic differentiation, as evidenced by metabolic activity, alkaline phosphatase activity, and calcium deposition.	[120]
BMP-2	CA/SF/PEO/nHA	Physical entrapment/Adsorption.	Rapid burst release in the first 3 days.	In vitro: The scaffold had a more profound effect on the attachment, proliferation, and osteogenic differentiation of BMSCs. In vivo: Enhanced bone regeneration in vivo.	[121]
BMP-2	HA/PG	nucleophilic affinity of oxidized PG and electrostatic interactions between inorganic particles incorporate the BMP-2.	Extended sustainable release.	In vitro: Enhanced osteogenic differentiation of hADSCs. In vivo: Improved bone formation in a calvarial bone defect.	[128]
BMP-2	Hyaluronan hydrogel with 25% HA	Chemically cross-linked.	Rapid burst release between day 3 and 2 weeks, followed by a slower release during the following 2 weeks.	In vivo: Improved bone formation in a rat bone defect.	[129]
BMP-2	SF/CS/agarose/HA with or without bioactive glass	-	Extended sustainable release.	In vitro: Increase the ALP activity in MC3T3 preosteoblast cells. In vivo: Promote bone formation in an ectopic muscle pouch model.	[130]
BMP-2	PELA/HA	-	A slow and sustained release.	In vivo: BMP-2 loaded can repair 5 mm bone defects in rat femora by 12 weeks.	[131]
BMP-2	nHA	-	-	In vitro: HA-1:1 model (means ridge vs. groove = 1:1) possessed excellent ability to capture BMP-2, less conformation change, and high cysteine-knot stability.	[132]
BMP-2	HAs HA HA-Pol	Physical entrapment/Adsorption.	-	In vitro: HA-Pol surface possessed high mass-uptake of rhBMP-2. ALP activity and Smad signaling increased in the order of HA-Sin < HA < HA-Pol. BMP-2 anchored on the HA-Pol surface with a relative loosened conformation, while the HA-Sin surface induced a compact conformation of BMP-2. rhBMP-2-adsorbed HAs possess a high cellular affinity.	[133]
BMP-2	HA/COL	Physical entrapment/Adsorption.	-	In vivo: BMP-2 can enhance bone formation in a canine sinus model.	[134]
BMP-2	PCL/PVAc/PLGA/nHA	Physical entrapment/Adsorption.	Rapid burst release.	In vitro: Promote cell proliferation and viability. Enhance the expression of ALP, OCN, OPN.	[135]
BMP-2	nHA/PLA/Gel	Utilize a polydopamine (pDA)-assisted coating to immobilize the BMP-2.	Sustained release.	In vitro: Elevated the ALP activity. Promoted the expression levels of RUNX-2, OCN, COL I. In vivo: BMP-2 can enhance bone formation in a rat cranial bone defect model.	[136]
BMP-2	PLLA/HA	Covalent binding.	Sustained release.	In vitro: Enhance the growth and osteogenic differentiation of MSCs. In vivo: Ectopic bone formation model exhibited significant bone formation.	[137]
BMP-2	SF/PEO/nHA	Physical entrapment/Adsorption.	A burst release.	In vitro: Promote the attachment, proliferation, and osteogenic differentiation of BMSCs. In vivo: Enhanced bone regeneration at 12 weeks post-implantation.	[138]
BMP-2	SF/nHA	embedded with SF microspheres.	A burst release on the first day.	In vitro: Promoted the adhesion and osteogenic differentiation of BMSCs. In vivo: Leading to complete bone bridging of rat cranial defects after 12 weeks of implantation.	[122]
BMP-2	GEF	In situ UV-crosslinking	-	In vitro: Enhanced osteo-differentiation of MSCs.	[138]
BMP9	COL1/HA/TCP	Recombinant adenovirus transfect C2C12 cells to express BMP9.	-	In vitro: Induced osteogenic differentiation in C2C12 cells. Induced robust and mature cancellous bone formation.	[125]
IGF	PLGA/HA	Covalent binding.	-	In vitro: Enhanced ADSCs attachment and proliferation. Increased ALP activity and expression of osteogenesis-related genes of ADSCs.	[139]
SDF-1	HA	Physical entrapment/adsorption.	Sustained release.	In vitro: Stimulated the migration of MSCs to the deep interior of the scaffold. Facilitated osteogenic differentiation. In vivo: promote the formation of blood vessels and bone.	[140]
VEGF	PCL/HA	rBMSCs were cultured in 50 ng/mL VEGF for 1 week.	-	In vitro: VEGF could enhance proliferation of rBMSCs. VEGF could promote the protein and mRNA expression levels of osteoblast- and endothelial cell related markers, such as OPN,COL 1,RUNX 2, VEGF, vWF, CD31. In vivo: VEGF could support the formation of vessels and bone.	[141]
VEGF	Mg/CDHA	Covalent binding.	A burst release in the first day.	In vitro: Improve the adhesion and proliferation of MC3T3-E1 cells. Promote the differentiation of rat MSCs into endothelial cells.	[142]
Ang2	HA/COL	Physical entrapment/adsorption.	-	In vivo: Enhance the expression of LC3 -I/LC3-II, Beclin-1, VEGF, and CD31. The new callus grew well, accompanied by remarkable angiogenesis and osteogenesis.	[143]
PDGF-BB	PLGA/nHA	Lentiviral vectors (LV-pdgfb) were physically entrapped.	Continuous release of lentiviral vectors (LV-pdgfb).	In vitro: PDGF-BB produced from BMSCs-P can enhance the migration of BMSCs. In vivo: The expression of pdgfb and the angiogenesis-related genes vWF and VEGFR2 were significantly increased. PDGF-BB facilitate angiogenesis and promote bone regeneration.	[144]
TGF-β1 BMP-2	mineral-coated HA microparticles	pTGF-β1 and pBMP-2 are physically entrapped.	-	In vitro: Binding efficiency of pDNA complex to MCM was quantified as 76.5%. MCM loaded with pTGF-β1 and/or pBMP-2 incorporated into cell aggregates can transfect MSCs and induce production of TGF-β1 and/or BMP-2, respectively. TGF-β1 production was sustained within day 4. BMP-2 production was sustained between day 12 to day 16. In vitro and in vivo: Delivery of TGF-β1 and pBMP-2 resulted in chondrogenesis and osteogenesis. MSCs exist donor-to-donor variability in differentiation capacity.	[145]
BMP-2 TGF-β	NSF/PCL/nHA	Carbodiimide coupling was used to load the growth factors.	A burst release over the first 3 days for both growth factors.	In vitro: Both growth factors can support cell adhesion, viability, and proliferation. Both growth factors can promote the expression of OPN, BSP, osteonectin, and collagen I, while BMP-2 can elevate the expression of Runx2, OCN, and ALP. Boost differentiation among the seeded osteogenic cells (evidenced by ALP activity and ARS staining).	[146]
TGF-β1 BMP-2	Gel microparticles and mineral-coated HA microparticles	TGF-β1 carried by Gel microparticles, while BMP-2 carried by HA microparticles.	Relatively rapid release of TGF-β1. More sustained release of BMP-2.	In vitro: TGF-β1 and BMP-2 loaded microparticles exhibited enhanced chondrogenesis and ALP activity. Staining for types I and II collagen, osteopontin, and osteocalcin revealed the presence of cartilage and bone.	[147]
TGF BMP-2	-	Physical entrapment/adsorption.	-	In vivo: TGF-β1 could induce cartilage formation in the early stage and BMP-2 promote remodeling into bone. Elevated the expression of types I, II, and X collagen, suggesting defect healing via endochondral ossification.	[148]
SDF-1 BMP-2	SF/nHA	SDF-1 is incorporated via physical adsorption BMP-2 is loaded into microspheres	Burst release of SDF-1 Sequential release of BMP-2.	In vitro and in vivo: SDF-1 can promote cell migration. BMP-2 can support cell proliferation and osteogenesis.	[149]
BMP-2 MEL	PLGA/CS/HA	Encapsulated in PLGA microparticles.	Sustainable release.	In vitro: Enhanced the proliferation of cells and the expressions of differentiation markers (RUNX2, ALP).	[150]
BMP-2 OGP	PLGA/HA	BMP-2 was carried by pore-closing process and layer-by-layer (LbL) assembly technique OGP was carried by PLGA microspheres.	Sustainable and Spatio-temporal release.	In vitro: Induce BMSCs osteogenic differentiation. Enhance ALP activity and osteogenic gene and protein (Runx2, COL I, and OCN) expression. BMP-2 makes contributions to osteogenic differentiation in an early stage while OGP accelerated proliferation and maturation of osteoblast precursors at a later stage. In vivo: Dual biofactor-loaded scaffold manifested the best repair efficacy.	[151]
BMP-2 VEGF	PCL/HA	Two growth factors were incorporated employing a multilayered coating based on polydopamine (PDA).	Sustainable and Spatio-temporal release.	In vitro: Enhanced proliferation of hEPCs and hMSCs. Upregulation of osteogenic (RUNX-2, ALP and OCN) and angiogenic (VEGF-A, VEGF-R2 and EDH-1) markers.	[152]
VEGF BMP-2	HA/COL	O-CMCS-O-Carboxymethyl chitosan microspheres were used to carrier rhBMP-2 and VEGF.	A burst release of VEGF, while a sustained release of BMP-2 was observed.	In vivo: rhBMP-2 could promote the formation of bone, while VEGF could efficiently promote the production and development of new blood vessels.	[153]
BMP-2 DFO	SNF/nHA	Physical entrapment/adsorption.	Extended sustainable release.	In vitro: Double loading of DFO and BMP-2 promote vascularization optimization. DFO and BMP-2 promoted ALP activity and the expression of levels of RUNX-2, OCN, OPN. In vivo: Improved new bone quality with both osteogenic and angiogenic features.	[34]
NG CGRP	HA/SA	Physical entrapment/adsorption.	-	In vitro and in vivo: Active proliferation and differentiation of BMSCs. Support the formation of bone.	[126]
CGF	nHA/COL	Physical entrapment/adsorption.	-	In vivo: Accelerate the rate of new bone formation. Increased the expression of OCN. Increased the BMP2 at 8 weeks. Elevate the compressive strength and elastic modulus.	[154]

PLL-poly(L-lysine); PDA-poly dopamine; CA-cellulose acetate, PG-pyrogallol; PELA-poly(d,l-lactic acid)-co-poly(ethylene glycol)-co-poly(d,l-lactic acid); HAs-hydroxyapatite surfaces; HA-Pol-crystal-coated polished surface; PCL-Polycaprolactone; PVAc-polyvinyl acetate; PEO-poly (ethylene oxide); GEF-gelatin electrospun fibrous; CDHA-Ca-deficient hydroxyapatite; Ang-2-Angiogenesis 2; NSF-Non-mulberry silk fibro; MEL -melatonin; OGP-osteogenic growth peptide; PCL-poly(ε--caprolactone); DFO-deferoxamine, small molecule compound; SNF-silk nanofibers; NG-naringenin; CGF-concentrated growth factor; Ref.-Reference.

**Table 4 materials-15-08475-t004:** The name, character, and composition of some HA composite products.

Products	Character	Composition	Porosity	Disadvantages	Ref.
TricOs T^®^	Synthetic complex	60% (*m*/*m*) HA, 40% (*m*/*m*) β-TCP; fibrin matrix	80%	NA	[206,207]
NanoBone^®^	Synthetic complex	Nanocrystalline HA embedded in silica matrix	61%	NA	[208,209]
Straumann^®^Boneceramic	Synthetic complex	60% (*m*/*m*) HA, 40% (*m*/*m*) β-TCP	90%	When the BoneCeramic^®^ was used as the preservation for the alveolar bone, the new bone formation contained connective tissue and less bone, which may impair the stability of the implant.	[210,211,212,213]
Maxresorb^®^	Synthetic complex	60% (*m*/*m*) HA, 40% (*m*/*m*) β-TCP	67.5%	NA	[210]
MBCP^®^	Synthetic complex	60% (*m*/*m*) HA, 40% (*m*/*m*) β-TCP	58%	NA	[214]
PBCP^®^	Synthetic complex	20% (*m*/*m*) HA, 80% (*m*/*m*) β-TCP	76%	NA	[214]
OsteoFlux^®^	Synthetic complex	TCP/HA	50–65%	NA	[215]
TCH^®^	Synthetic complex	-	60.3–63.7%	NA	[216]
CopiOs^®^	Xenograft substitutes	-	65.3–82.4%	NA	[216]
NuOss^®^	Xenograft substitutes	-	80%	NA	[217]
Osteobiol^®^	Xenograft substitutes	-	33.1%	NA	[217]
Lubboc^®^	Xenograft substitutes	-	67.3–82.5%	NA	[216]
Smartbone^®^	Xenograft substitutes	-	27%	NA	[217]
Cerabone^®^	Xenograft substitutes	-	69.0%	NA	[208,210,218]
Maxgraft^®^	Allograft substitutes	-	-	NA	[208]
Osteopure^®^	Allograft substitutes	-	76.7–82.2%	NA	[216]
Puros^®^	Allograft substitutes	-	-	NA	[219]

Ref-reference; NA—not available.

## Data Availability

Not applicable.

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
