# Peer review of "Frontiers of Hydroxyapatite Composites in Bionic Bone Tissue Engineering"

_materials, 2022, doi:10.3390/ma15238475_

Round 1

Reviewer 1 Report

The authors have prepared a comprehensive review of Hydroxyapatite Composites materials used in bone defects.

·         Abstract is quite adequate and gives a concise appearance to the whole work.

·         It is advised that the writers discuss bone healing processes in the introduction and include details, particularly radiographic information.

·         The general work flow is properly and consistently organized.

·         I think that the language used is sufficient and there is no need for proofread.

·         The used figures are of sufficient quality.

·         But even so, the reference list's currentness is more crucial. It is assumed that the authors paid attention to the requirement that references utilized in these studies be current.

Author Response

1.It is advised that the writers discuss bone healing processes in the introduction and include details, particularly radiographic information.

Reply: We discussed the bone healing processes and radiographic information in the part of introduction (page 2,3 line 62-83).

2.The reference list's currentness is more crucial. It is assumed that the authors paid attention to the requirement that references utilized in these studies be current.

Reply: Thank you for the important suggestions. We further paid attention to the reference list's currentness, and renewed the current studies of HA in the revised manuscript. The publish time of references utilized in the manuscript are shown as follows. Studies published in the recent 3 years (2020-2022) make up for 30.45%, and 53.64% articles were published in the recent 5 years (2018-2022).

Publish time of article

Number of article

others

31

2014

20

2015

14

2016

20

2017

17

2018

26

2019

25

2020

19

2021

40

2022

8

Reviewer 2 Report

The paper “Frontiers of Hydroxyapatite Composites in Bionic Bone Tissue 2 Engineering” makes a very extensive and complete review on the different HA materials developed for bone repair applications. The review is well structured and written. A few mistakes can be found like the one in page 8 line 238 “added poly (glycolic acid) 236 (PGA)” I believe that it should be “added poly glycolic acid 236 (PGA)”. Thus, it is important to make a revision of the whole text to avoid this kind of mistakes. The paper is easy to read and highlights the best results in different areas studied for the developed materials base on Ha and its composites. I just have one major question o requirement; the authors should make deeper conclusions by establishing the best conditions and limitations of such materials as bone repair implants. For example: when is better to use HA materials instead of metallic scaffolds. What is the next steps to improve the disadvantages of such materials, etc..

Congratulations to the authors that did a great job!

Author Response

1.A few mistakes can be found like the one in page 8 line 238 “added poly (glycolic acid) 236 (PGA)” I believe that it should be “added poly glycolic acid 236 (PGA)”

Reply: Thank you for the important suggestion. Poly (glycolic acid) stands for poly-glycolic acid. Similar nouns may appear without parentheses, but it is actually inaccurate and can sometimes lead to confusion about the structure of the polymer.

2.The authors should make deeper conclusions by establishing the best conditions and limitations of such materials as bone repair implants. For example: when is better to use HA materials instead of metallic scaffolds. What is the next steps to improve the disadvantages of such materials, etc.

Reply: We discussed the advantages and disadvantages of the scaffolds commonly used in the bone tissue engineering in the introduction (page 3, line 89-104). Besides, we emphasized areas that need further studies in the end of each section and the part of conclusion (page 9 line 214-217, page 12 line 323-329, page 30 line 696-703).

Reviewer 3 Report

Review for Manuscript ID: materials-2011715 entitled " Frontiers of Hydroxyapatite Composites in Bionic Bone Tissue Engineering”

The manuscript is of interest and has merit for publication. However, there are points that need to be corrected as follows: 

1-    Abstract, the conclusions need to be added. 

2-    Figure 1: abbreviations need to be defined in the legend. 

3-    Line 72-76: better to be in line 64 after the citation of figure 2.

4-    I think it is better for each section to have a clear message at the end of that section to allow the reader to know the impact of each of the parameters on HA scaffold. 

5-    I think in the conclusions, the reader should know the ideal characteristics of HA scaffold according to the literature for each variables mentioned. Furthermore, the author should highlight the area that need further studies. 

6-    It is also necessary to have a table comparing HA scaffold to other types of scaffolds. This table should show why HA is superior to other scaffolds. 

BW, 

Author Response

1.Abstract, the conclusions need to be added. 

Reply: We add the conclusions at the end of the abstract (page 1 line36-38).

2.Figure 1: abbreviations need to be defined in the legend. 

Reply: We defined the abbreviations of BMMSCs and BMP2 in the legend of Figure 1 (page 2 line 61).

3.Line 72-76: better to be in line 64 after the citation of figure 2.

Reply: Line 72-76 “Scaffold materials should be biocompatible, that is, non-toxic and harmless to cells; Force conductivity; Degradability; Porous, used to contain cells and cytokines, and can also be used for the discharge of nutrients and metabolic waste; Mechanical compatibility, that is, it has similar mechanical strength and elastic modulus with natural bone; It can load seed cells, cytokines or drugs.” was the legend of Figure 2.

4.I think it is better for each section to have a clear message at the end of that section to allow the reader to know the impact of each of the parameters on HA scaffold. 

Reply: The significance of each parameters are highlighted in each section (page 6 line173-177, page 9 line210-214 221-222, page 10 line 238-246, page 11 line 267-274 278-279, page 12 line 322-323, page 14 line 341-343, page 22 line 437-440, page 25 line 564-565, page 27 line 632-635)

5.I think in the conclusions, the reader should know the ideal characteristics of HA scaffold according to the literature for each variables mentioned. Furthermore, the author should highlight the area that need further studies. 

Reply: The ideal characteristics of HA scaffold according to the literature are statement in the part of conclusion (page 29 line 690-695). Besides, we emphasized areas that need further studies in the end of each section and the part of conclusion (page 9 line 214-217, page 12 line 323-329, page 30 line 696-703).

6.It is also necessary to have a table comparing HA scaffold to other types of scaffolds. This table should show why HA is superior to other scaffolds. 

Reply: We discussed the advantages and disadvantages of the scaffolds commonly used in the bone tissue engineering in the introduction (page 3, line 89-104). HA process strong bone-inducing response and good biocompatibility. However, it also exists the shortcoming of long period of degradation and high brittleness, which limits its application. Composite scaffolds made of HA and other materials can avoid the shortcomings of simple HA scaffolds and have better performance, which is also the main content of this review.